# Amino Acids in Entomopathogenic Fungi Cultured In Vitro

**Lech Wojciech Szajdak \*, Stanisław Bałazy and Teresa Meysner**

Institute for Agricultural and Forest Environment, Polish Academy of Sciences, ul. Bukowska 19, 60-809 Poznań, Poland; Stanislaw.balazy@isrl.poznan.pl (S.B.); teresa_meysner@tlen.pl (T.M.)

\* Correspondence: lech.szajdak@isrl.poznan.pl; Tel.: +48-61-8475603

**Abstract:** The content of bounded amino acids in six entomopathogenic fungi was identified and determined. Analyzing the elements characterizing the pathogenicity of individual species of fungi based on infectivity criteria, ranges of infected hosts, and the ability to induce epizootics, these can be ranked in the following order: *Isaria farinosa, Isaria tenuipes, Isaria fumosorose, Lecanicillium lecanii, Conidiobolus coronatus, Isaria coleopterorum.* These fungi represent two types of *Hyphomycetales-Paecilomyces* Bainier and *Verticillium Nees* ex Fr. and one type of *Entomophtorales-Conidiobolus* Brefeld. Our study indicates that there are significant quantitative and qualitative differences of bounded amino acids in the entomopathogenic fungal strains contained in the mycelium between high and low pathogenicity strains. The richest composition of bounded amino acids has been shown in the mycelium of the *Isaria farinosa* strain, which is one of the most commonly presented pathogenic fungi in this group with a very wide range of infected hosts and is the most frequently recorded in nature as an important factor limiting the population of insects.

**Keywords:** bounded amino acids; entomopathogenic fungi; strain pathogenicity

---

## 1. Introduction

Plant pathogens created by insects significantly reduce crop yields. These pathogens strongly impact the epidemiology of a plant disease. If we want to decrease use of pesticides, then we should offer alternative ways to resolve the problem of pest damage. The choice of an effective method depends on our knowledge of the pests' biology and their behavior. The aim is to manipulate pests' physiology to decline the damage they cause. Therefore, the knowledge of insect ecology should help us identify the processes and mechanisms of insect response, which help us to propose new methods and tools that effectively decrease that response.

The use of entomopathogenic fungi as biocontrol agents instead of pesticides is an attractive and an alternative method, which can efficiently control the pest population. In addition, entomopatheogenic fungi contrary to pesticides do not have any negative hazard effect on human health and environment. Therefore, fungal pathogens of arthropods have been generally considered as beneficial organisms for agriculture and forestry because of their ability to significantly decrease the populations of many economically important pest insects and mites. Their numerous strains have been included into the integrated pest management programs [1], which offer the opportunity to eliminate or drastically reduce the use of pesticides and to minimize the toxicity of an exposure to any chemicals which are used.

World-wide, the integrated pest management programs have become the suggested strategy for plant protection. Most programs for this strategy suggest that the efficient assessment of the importance of diversity for significantly decreasing pest population fluctuation has been used for the establishment of various crops, with consequent comparison of the occurrence and multiplicity of pest

populations and their enemies. These programs have linked to broader ecology of pest management and lead to an avoidance of toxic chemicals and other developmental disorders.

These programs are related to human health, environmental, and economic issues of crop yield addressed to significantly control pest populations through a variety of biotic/abiotic technologies. The main aim of these technologies is related to preventing unacceptable levels of pest devastation, while minimizing risk to human health, depletion of resources, and degradation of ecosystems.

For these technologies, preservation of the richest possible genetic pool of those pathogens in the natural environment has the greatest significance. Pathogenic fungi are mostly intracellular pathogens, indicating that at some point during the interaction between the host and the invading species, the pathogen lives inside the host cell. Long-term studies in various European habitats have shown the strong impoverishment of these pathogenic fungi in rural areas compared with natural reserves, forest ecosystems, cultivation and production of horticultural crops [2–5].

The formation of differentiated landscape structure of arable fields, abounding in small non-productive natural elements (shelterbelts, woodlots, bushes, wetlands, semi natural plant association, ponds, water bodies, and others) favors the preservation of a greater diversity of entomopathogenic fungi. On the other hand, the majority of encountered entomopathogenic fungi show the ability to infect or damage both noxious and beneficial arthropods, whereas some fungal strains can affect only the latter [6,7].

The study into the physiology and biochemistry of fungi has so far provided a large amount of data on bounded amino acids found in the thalluses but this information is still unsatisfactory [8,9]. Amino acids are an integral part of the hydrophilic-hydrophobic properties of fungal cell walls and they are included in the structure of organic substances produced by these microorganisms [10,11].

In the group of entomopathogenic fungi, bounded amino acids are recognized relatively weakly in the composition of metabolites, enzymes and toxins occurring more commonly and widely in the following species *Hyphomycetales-Beauveria bassiana* (Bels.) Vuill., *Beauveria brongniartii* (Sacc.) Petch, *Metarhizium anisopliae* (Metschnikof) Sorokin, *Paecilomyces fumosoroseus* (Wize) Brown et Smith, *Paecilomyces lilacinus* (Thom) Samson, *Verticilium lecanii* (Zimmerman) Viegas and in *Cordyceps* spp., *Entomophthorales* [10–20].

The aim of the preliminary study was to identify the bounded amino acids composition produced by several poorly studied and known entomopathogenic strains of *Hyphomycetales* and one *Entomophthorales* under culture conditions on a mineral medium with saccharose and try to find the relationship between the content of bounded amino acids and pathogenicity strain.

## 2. Materials and Methods

### 2.1. Materials

The major taxonomical units comprising entomopathogenic species are the orders of *Blastocladiales* and *Entomophthorales* classified in the lower fungi and among higher fungi, some genera of the order *Hypocreales* (*Cordyceps, Hypocrella, Torubiella*) often associated with their anamorphs grouped in pycnidial or synnematous *Deuteromycetes* (*Akanhomyces, Aschersonia, Gibellula, Hirsutela, Paecilomyces* and others) and a great number of conidial forms whose relations to perfect stages remain still unknown or vague (*Aphanocladium, Beauveria, Paecilomyces, Sorosporella, Verticillium* and others). In our study, the following six species of the entomopathogenic fungi (*Isaria farinosa, Isaria tenuipes, Isaria fumosorosea, Lecanicillium lecanii, Conidiobolus coronatus* and *Isaria coleopterorum)* representing two types of *Hyphomycetales-Paecilomyces* Bainier and *Verticillium Nees* ex Fr. and one type of *Entomophtorales-Conidiobolus* Brefeld were evaluated (Figure 1).

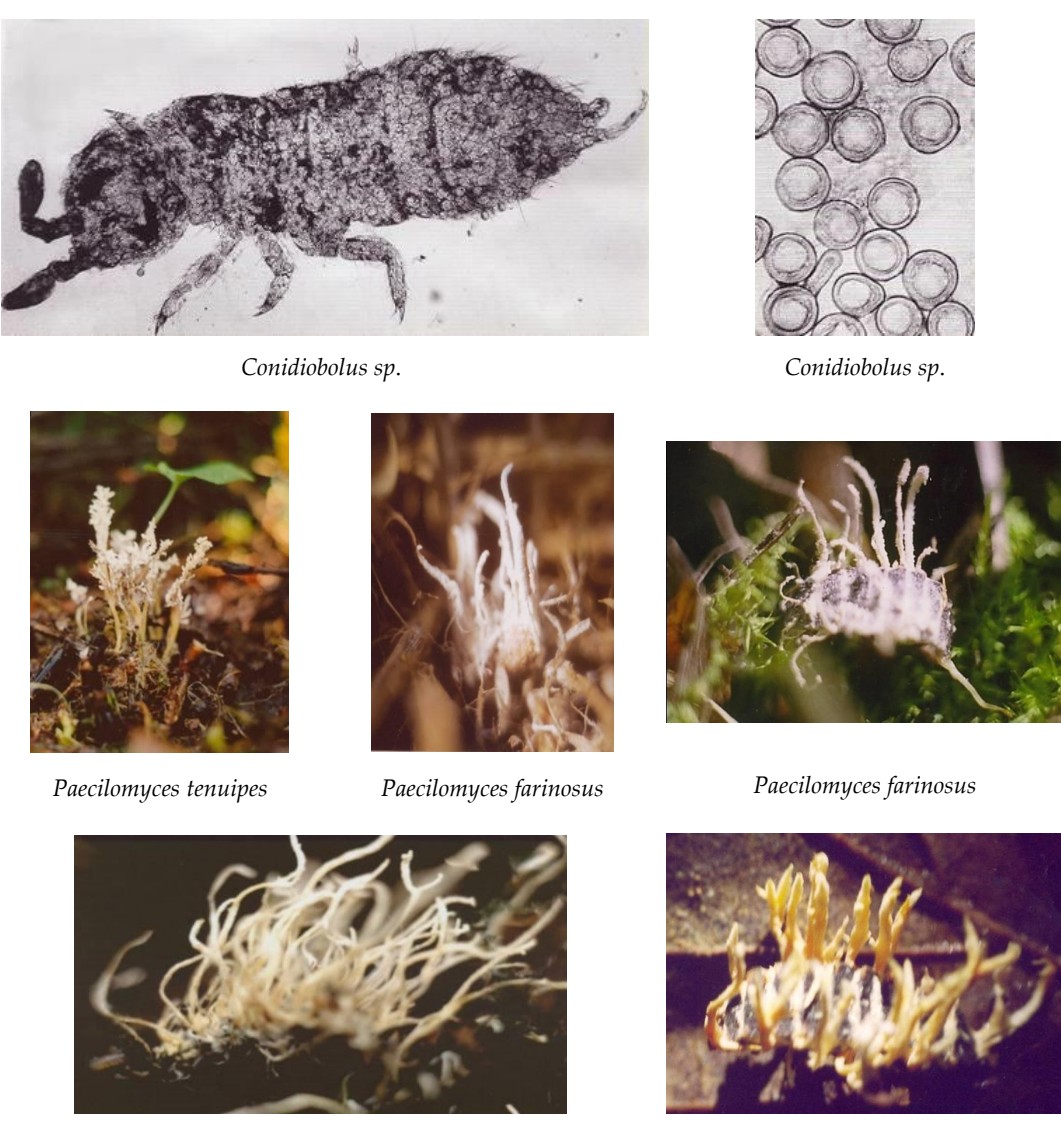

**Figure 1.** Entomopathogenic fungi (private collection of prof. Stanisław Bałazy).

### 2.1.1. Fungi Characterisation

1. *Isaria farinosa* (synonyms: Ramaria farinosa, Clavaria farinosa, Corynoides farinosa, Spicaria farinosa, Penicillium farinosum, Paecilomyces farinosus, Isaria psychidae, Penicillium alboaurantium) (MycoBank) was isolated from an unspecified insect species collected in the forest litter of the Wielkopolski National Park (Poland). This strain is characterized by growth and sporulation on commonly used mineral-organic media in saprophyte breeding, showing high pathogenicity and intensive development during the entire vegetative season, beginning from early spring to late autumn in meadows and forests [21]. This fungus belongs to the polyphagous species and causes severe epizootics in butterflies of larvae and pupae. Isaria farinosa is one of the most common species of entomopathogenic fungi with a worldwide distribution and a comparatively wide host range [22].

2. *Isaria tenuipes* (synonyms: *Paecilomyces tenuipes, Spicaria heliothis*) (MycoBank) was collected from an undetermined insect species and isolated near *Vladivostok, Primorsky Krai* (Russian Federation) and obtained from Dr. V.A. Borisova. In terms of morphology, it does not correspond to any of the species described so far, but *Paecilomyces amoenoroseus* (P. Henn.) Samson is very close. The range of the hosts is unrecognized. In laboratory conditions, *Isaria tenuipes* infects 100% of bark beetles, saprophagic and

hymenoptera (*Hymenoptera*) larvae from *Parasitica* group. Development and sporulation in vitro is very fast and intensive on various media.

　　3. Isaria fumosorosea (synonyms: Spicaria fumosorosea, Paecilomyces fumosoroseus, Spicaria aphodii, Monilia aquatilis, Paecilomyces hibernicus, Paecilomyces hibernicum, Paecilomyces isarioides) (MycoBank) was collected from soil under the spruce stand in the floodplain forest, Strass near Laufen (Bawaria, Germany). This species is considered polyphaga but with a much narrower host range than Isaria farinosa. This species is commonly found in the soil [23], but cases of insect infections are encountered less frequently and usually locally. Moreover this fungus has been reported on plants, in water and less commonly in the air on every continent except Antarctica [4]. Some of the more commonly known susceptible organisms include weevils, ground beetles, plant beetles, aphids, whiteflies, psyllids, wasps, termites, thrips and a wide variety of butterflies and moths [24,25]. The development takes place in vitro.

　　4. Lecanicillium lecanii (synonyms: Cephalosporium lecanii, Verticillium lecanii, Actostalagmus albus-minimus, Actostalagmus coccidicola, Cephalosporium coccorum) (MycoBank) was isolated with imago Aleurodidae in Wielkopolski National Park (Poland). Intensive development and sporulation was observed on media. This species is highly pathogenic to beetle larvae, dipterans and hymenoptera in laboratory conditions. Endophytic colonization has been reported by Ownley [26] for Lecanicillium lecanii and it has been suggested that induced systemic resistance may be active against powdery mildew. Mycoparasitism is the primary mechanism employed by this fungus against plant pathogens. These strains are considered to be polyphagous with broad ranges of potential hosts, but they require high humidity for the initiation of epizootics.

　　5. *Conidiobolus coronatus* (Constantin) Batko (synonyms: *Boudierella coronata, Delacroixia coronata, Entomophthora coronata, Conidiobolus villosus*) (MycoBank) was isolated from the soil of the shelterbelt near Sains in Normandia (France). The sporulation is intensive on all culture media but prefers the substrates enriched with peptone or yolk. *Conidiobolus coronatus* is a saprophytic species of entomopathogenic fungus, with a tendency to parasitism. This species is characterized by low host selectivity and strong post-infection aggressiveness caused by highly toxic metabolites of invertebrates [27]. This fungus is the opportunistic pathogen of a broad range of infected hosts eg. *Galleria mellonella, Dendrolimus pini* [28]. This species was also noted on mushrooms, herbaceous plants, vertebrates and individual cases on aphids [29]. Under experimental conditions, the test strain showed pathogenic properties on larvae of *Tribolium destructor Uytenb.* (*Tenebrionidae*) but only 30% of positive infections were in conditions of 80% moisture. Recent study showed that this fungus is the common pathogen of soil microflora and lives mainly on the edges of forests, meadows, gardens, and parks [27,30].

　　6. *Isaria coleopterorum* (synonyms: *Paecilomyces coleopterorum*) (MycoBank) was isolated from beetle larvae of the family *Cantharidae* (*Lampyris* sp.) collected on forest in Moosen near Laufen (Bavaria, Germany). Cultural characteristics and morphology of the mycelium and spores do not correspond to any of the previously known species of the genus *Paecilomyces*. This fungus grows well on the mineral-organic media, but better grows abundantly and sporulates efficiently on media enriched protein substances (peptone, broth, egg yolk). The host range is unknown. Laboratory sample infection of several species larvae of beetles, flies and *Hymenoptera* proved unsuccessful.

　　Soil samples were collected in triplicate from the field (see fungi characterization) and stored in plastic bags at the temperature of 3–4 °C. Immediately before starting the experiment in the laboratory, the soil was sieved (1 mm sieve) to separate the larger particles of impurities and dried up to a moisture content of approximately 25–30% (which is optimal for fungal growth and limits the growth of entomopathogenic nematodes) [31]. The cultures of the fungi used for the bioassays and bounded amino acids study were deposited in the collection of the Institute for Agricultural and Forest Environment, Polish Academy of Sciences in Poznań, Poland.

　　In isolated fungi, the disease agents, their frequency, duration and succession were identified by common insect pathology techniques [32]. Each fungal suspension contained $1.0 \times 10^9$ spores of fungi in 1 mL. This volume was taken and the sporulation fungi from cadavers grown on sterilized Czapek's

agar: saccharose, sodium nitrate, dipotassium phosphate, magnesium sulfate, potassium chloride, ferrous sulfate, agar and demineralized water at pH = 7.3.

In order to obtain cultures, small portions of external mycelium from host cadavers were inoculated on the surface of the agar. They were allowed to grow on the Petri dishes (9 cm diameter) for 5 days on a rotary shaker (90 rpm) at 20 ± 1 °C. The cultures were washed in sterile redistilled water three times (wash bottle, gentle), filtered (paper Whatman GT/C), and dried at 80 °C for 4 h. For taxonomical differentiation of fungal strains, the macro- and micro-morphological features of the mycelium and sporulation on the host mites and in cultures were compared according to the method of Bałazy et al. [33].

### 2.1.2. Reagents

(a). Amino acids standards: Amino acids were purchased from Sigma-Aldrich (St. Lous, MO, USA). The concentration of each amino acid is equal to 2.5 µM in 1 mL solution (Table 1), (Figure 2) however, the quantity of cystine is one-half of the amounts. The standards solutions prepared commercially should always keep in the refrigerator. The manufacturer informs the concentrations of every amino acid in standard solution (Table 1).

(b). Lithium citrate elution buffers

(c). 4 N acetate buffer at pH = 5.5. Dissolve 1.0880 g sodium acetate in 800 mL boiling deionized water. Decrease the solution to laboratory temperature and add 200 mL of ice acetic acid.

(d). Ninhydrin solution: It must be prepared in black bottle. Mix 1500 mL of methyl cellosolve $C_3H_8O_2$ (ethylenglycol monomethyl ether) and 500 mL of 4 N acetate buffer. Add to this solution 40 g ninhydrin (analytical grade) and stir the solution under argon for dissolving ninhydrin. After dissolving ninhydrin, 0.8 g $SnCl_2$ $2H_2O$ is added and the stirring is done. Keep the solution of ninhydrin in refrigerator.

(e). Separation of amino acids: The glass column [0.37 × 20 cm], was packed with resin Ostion LGFA. Although, resin Ostion LGKS$^{0804}$ fills in the precolumn (1 × 8 cm). Both Ostion LGFA and Ostion LGKS$^{080}$ are strongly cation exchange resins.

The resin should be converted into the $H^+$ form. The process is carried out by 15% HCl (1:5 ratios) on the water bath at 80 °C for 1 h. The suspension has to be stirred and next filtered by filter paper Whatman GT/C and washed by deionized water until pH neutral.

Resin is converted from $H^+$ form to $Li^+$ form using 2 N LiOH. After washing, resin was mixed by the first elution buffer (pH = 2.9), (ratio 1:5) and allowed to stay overnight. Then the buffer is sucked off and the dilution 1:1 is left unchanged and the column is ready for filling.

Amino acids were separated with five lithium citrate buffers (Table 2). The buffers were prepared with water generated with Mili-Q water purification system (Millipore, Molsheim, France).

**Table 1.** Standard solution for the analyzing amino acids. Amounts of amino acids are in 1 L.

| Amino Acid | Amount Per 1 L (g) |
| --- | --- |
| Cysteic acid | 234 |
| Taurine | 156.4 |
| Phosphoetanoloamine | 176.38 |
| Aspargig acid | 166.37 |
| Hydroxyproline | 163.9 |
| Threonine | 148.87 |
| Serine | 131.37 |
| Asparagine hydrate | 187.7 |
| Glutamic acd | 184 |
| Glutamine | 182.7 |
| α-amino adipic acid | 201.5 |
| Proline | 143.87 |
| Glycine | 93.87 |
| Alanine | 111.38 |
| Citruline | 219 |
| α-aminobutyric acid | 128.9 |
| Valine | 146.37 |
| Cystine | 150.19 |
| Methionine | 186.5 |
| Cystationine | 277.86 |
| Isoleucine | 164 |
| Leucine | 164 |
| Norleucine | 131.2 |
| Tyrosine | 226.5 |
| Phenylalanine | 203.13 |
| β-alanine | 111.4 |
| β-aminoisobutyric acid | 128.9 |
| γ-aminobutyric acid | 128.9 |
| Ornithine | 210.75 |
| Lysine | 228.37 |
| 1-methyl histidine | 234 |
| 3-methyl histidine | 211.5 |
| Tryptophane | 255.3 |
| Arginine | 217.8 |

**Table 2.** Lithium citrate elution buffers per 1 L.

| | |
| --- | --- |
| 1 | *Lithium citrate buffer at pH = 2.9:* 9.4000 citric acid, 9.3880 g lithium chloride, 2.3000 g lithium citrate, 2 mL 30% Brij 35®, 0.1 mL caprylic acid, 2.5 mL thiodiglycol (Pierce Chemical Co., Dallas, TX, USA) |
| 2 | *Lithium citrate buffer at pH = 3.1:* 9.7000 g citric acid, 9.5700 g lithium chloride, 3.9000 g lithium citrate 2 mL 30% Brij 35®, 0.1 mL caprylic acid, 2.5 mL tiodiglycol (Pierce Chemical Co., USA); |
| 3 | *Lithium citrate buffer at pH = 3.35:* 10.4980 g citric acid, 17.5000 g lithium chloride, 5.6500 g lithium citrate, 2 mL 30% Brij 35®, 0.1 mL caprylic acid, 2.5 mL tiodigliycol (Pierce Chemical Co., USA); |
| 4 | *Lithium citrate buffer at pH = 4.05:* 9.5000 g citric acid, 10.0000 g lithium chloride, 15.4500 g lithium citrate, 2 mL 30% Brij 35®, 0.1 mL caprylic acid, 2.5 mL tiodiglikolu (Pierce Chemical Co., USA); |
| 5 | *Lithium citrate buffer at pH = 4.9:* 8.5000 g citric acid, 39.9500 g lithium chloride, 50.6300 g lithium citrate, 2 mL 30% Brij 35®, 0.1 mL caprylic acid. |

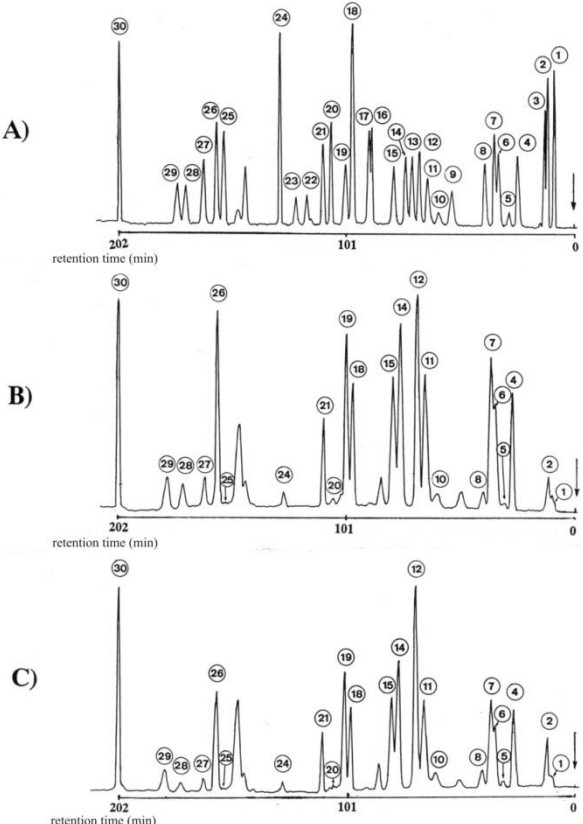

**Figure 2.** Chromatograms of amino acids: (**A**) standards, (**B**) hydrolysates from *Paecilomyces* sp. II, (**C**) hydrolysates from *Paecilomyces fumosoroseus*. Where: 1-cysteic acid, 2-taurine, 3-phosphoethanolamine, 4-aspartic acid, 5-hydroxyproline, 6-threonine, 7-serine, 8-glutamic acid, 9-α-aminoadipic acid, 10-proline, 11-glycyne, 12-alanine, 13-citrulline, 14-α-aminobutyric acid, 15-valine, 16-cysteine, 17-methionine, 18-cystathionine, 19-isoleucine, 20-leucine, 21-tyrosine, 22-β-alanine, 23-β-aminobutyric acid, 24-γ-aminobutyric acid, 25-ornithine, 26-lysine, 27-histidine, 28-1-methylohistidine, 29-3-methylohistidine, 30-arginine.

Determination of Bound Amino Acids in Hydrolysates Mycelium

*Calculations*

The determination of amino acids is based on the ninhydrin reaction:

The ninhidrin-$CO_2$ method is highly specific and used in detection and assay of amino acids. It requires a free $NH_2$ group adjacent to a -COOH group. The control of pH has high restriction for this determination. Ninhydrin creates $NH_3$ blue colored compound in higher pH value shown below.

This analysis is recommended for the material, where amino acids are in low amounts (body fluids, mineral and organic soils, feed etc.).

The retention time of each amino acid seems to be a characteristic value for its identification. It is possible to identify these amino acids and/or ninhydrin positive substances present in the calibration solution.

The method is characterized by high specificity. The differences between the retention times of amino acids in hydrolysates mycelium and in standard mixtures were below 2%, for wide range of concentrations the linear correlation coefficients ranged from 0.992 to 0.997, high accuracy (recovery 92.9–108.5%), the precision of the results below 4.2% in case of repeatability.

The column (resin Ostion LGFA) is designed for rapid, high-resolution amino acids separations. Column has high capacity and reproducibility, making them ideal for the intermediate and polishing steps of the laboratory-scale amino acids separation workflow. The connection of the column to the sequence of lithium-citrate buffers is responsible for the large number of separated amino acids.

*Precautions*

All buffers have to be adjusted precisely, however, ninhydrin solution should be aged at least 14 h. Flow rates of the buffers and ninhydrin as well as temperatures used for separation of amino acids have to be set. The standard solution must be prepared quantitatively, using those amino acids, which are not represented in commercial standards. The control of instrument has to be checked every day.

## 2.2. Method for the Separation and Determination of Amino Acids in Mycelium Hydrolysates

The concentrations of amino acids were determined in hydrolysates mycelium. To obtain hydrolysates mycelium, 20 mL 6 M HCl was added to 0.2 g of sample in the test tube. An amount of 6 N HCl was used to perform the hydrolysis of proteins and peptides. This chemical agent was prepared by diluting the concentrated HCl 1:1 distilled water. The tube was allowed to stay overnight. The next day, the test tube was put on the microwave water bath to remove gasses. Then argon was babbled for 10 min to remove oxygen. Tubes were closed tightly and heated for 24 h at 15 °C. Solutions from the test tubes were transferred to 50 mL flasks. The solution was evaporated to dry mass. Dry mass was dissolved in 0.6 mL ethyl alcohol (96%) and added to 0.4 mL 40% trichloroacetic acid and 1 mL deionized water. The solution was centrifuged and the supernatant was used for analysis.

Separation and determination of 30 bounded amino acids was carried out on T 339 Amino Acids Analyzer (Mikrotechna-Prague) (ion-exchange chromatography, the strong cation exchange resin) [34]. The separation of the bound amino acids was achieved by injecting 100 mL samples into an Ostion LGFA (0.37 × 20 cm) column. Lithium-citric buffers of the following pHs were used: 2.90, 3.10, 3.35, 4.05 and 4.90, and the absorbance of the eluent–ninhydrin complex was monitored at $\lambda_{max} = 520$ nm (Szajdak 1996). The mobile phase was pumped at 12 cm$^3$ h$^{-1}$ and developed a pressure of 2.5 MPa. A full-range recorder span of 100 mV was used to provide on-scale peaks. With this system, adequate resolution of a large number of amino acids up to and including arginine can be achieved in a chromatographic run of 4 h (total run time) (Figure 2).

Before the next sample injection, the column was equilibrated with buffer 1 for 20 min.

Chemicals were accurately weighed. Each buffer was filtered through a 0.22 μm filter (Millipore, Molsheim, France) before use, and kept under argon in the apparatus. Throughout the elution program, the flow rate for buffer solutions and ninhydrin was 12 mL/h. During the running of the samples, the separate pump system automatically mixes buffer and ninhydrin reagent, which are kept under argon in the instrument.

All chemical analyzes were run in triplicate and the results were averaged. The confidence intervals for the mean were calculated using the following formula: $\overline{X}t_{\alpha(n-1)}$ SE, where: $\overline{X}$—mean, $t_{\alpha(n-1)}$—value of the Student test for $\alpha = 0.05$, and n − 1 degree of freedom, Wpisz tutaj równanie. SE—standard error.

## 3. Results and Discussion

Infection begins with a connection of conidia or blastospores to the insect cuticle. A variety of hydrolytic enzymes, e.g., proteases, chitinases, and lipases and other factors, contribute to the germination and growth of the fungus across the surface of the host and penetration of cuticular layers. During this process the fungus produces any number of specialized infection structures that can include penetration pegs or appressoria, which enable the growing hyphae to penetrate into the host integument. It is essentially in the later stages of this process that the pathogen encounters the host immune system. The insect cuticle is a highly heterogeneous structure and can vary greatly in the composition even during the various life-stages of a particular insect. The epicuticle or outermost layer provides a hydrophobic barrier rich in amino acids in peptides and lipids and is followed by the procuticle that contains chitin and sclerotised protein [9,35–50]. Therefore, hydrophobic mycotoxins will be easily transported throughout this barrier and the knowledge about the content of bounded amino acids is needed.

The total amount of bounded amino acids in the mycelium of the individual fungal strains tested was clearly diverse. The highest total content of bounded amino acids (14,963.0 µg g$^{-1}$) was related to *Isaria tenuipes* (*Paecilomyces sp.II*), while the lowest (390.0 µg g$^{-1}$) was measured in mycelium of *Conidiobolus coronatus* (Table 3).

In the mycelium samples, the following groups of bounded amino acids were tested: sulfuric, acidic, neutral and basic (Table 4). Neutral amino acids dominated. Their contents in the individual strain of entomathogenic fungi ranged from 38 to 57% of all amino acids determined. However, the content of acidic amino acids ranged from 17 to 30% of the total amino acids identified. The lowest percentage content was found for basic amino acids which amounted to from 8 to 32%. The concentrations of sulfuric amino acids in all analyzed strains of fungi were the lowest among these groups and their contents ranged from 4 to 11%.

In *Isaria farinosa, Isaria tenuipes, Isaria fumosorosea, Leanicillium lecanii* a group of hydrophobic amino acids dominated and their content ranged from 53 to 66% (Table 5). On the other hand, hydrophilic amino acids were found to be from 34 to 47% in the tested entomathogenic fungi. With reference to *Conidiobolus coronatus* and *Isaria coleopterorum*, the dominant groups in the hydrolysates from their mycelium were generally hydrophilic amino acids, whose percentages were close to approximately 60% and hydrophobic amino acids at 40%, respectively.

From 21 to 24 amino acids were identified in various strains of the mycelium samples. This number is significantly longer than in published studies (Table 6).

Table 3. Content of bounded amino acids in entomopathogenic fungi in µg g$^{-1}$ d.m. of fungi, and in percent (italics).

| Amino Acids | Isaria farinosa | Isaria tenuipes | Isaria fumosorosea | Lecanicillium lecanii | Conidiobolus coronatus | Isaria coleopterorum |
|---|---|---|---|---|---|---|
| Cysteic acid | 67.6 ± 2.5 *1.3* | 32.4 ± 1.2, *0.2* | 27.4 ± 1.0 *0.3* | 14.6 ± 0.5 *0.2* | 24.8 ± 0.9 *6.4* | 17.3 ± 0.6 *2.5* |
| Taurine | 28.7 ± 1.0 *0.5* | 141.3 ± 5.3 *0.9* | 131.0 ± 4.9 *1.3* | 49.9 ± 1.6 *0.5* | 1.7 ± 0.1 *0.4* | 1.7 ± 0.1 *0.3* |
| Phosphoethanolamine | - | - | - | - | 37.9 ± 1.4 *9.7* | 37.8 ± 1.2 *5.5* |
| Aspargic acid | 32.0 ± 1.2 *0.6* | 757.9 ± 28.5 *5.1* | 558.0 ± 21.0 *5.5* | 441.5 ± 16.6 *4.7* | 15.8 ± 0.6 *4.1* | 45.5 ± 1.7 *6.7* |
| Hydroksyproline | 1196.9 ± 44.9 *22.5* | 322.4 ± 12.1 *2.2* | 110.5 ± 4.2 *1.1* | 139.9 ± 5.3 *2.1* | 30.7 ± 1.2 *7.9* | 46.0 ± 1.7 *6.7* |
| Threonine | - | 705.0 ± 26.5 *4.7* | 380.0 ± 14.3 *3.6* | 445.0 ± 16.7 *4.7* | 26.4 ± 1.0 *6.8* | 24.1 ± 0.9 *3.5* |
| Serine | 26.6 ± 1.0 *0.5* | 865 ± 32.6 *5.8* | 523.21 ± 9.7 *4.9* | 530.21 ± 9.9 *5.7* | 5.7 ± 0.2 *1.5* | 21.0 ± 0.8 *3.1* |
| Glutamic acid | 294.4 ± 11.0 *5.5* | 194.4 ± 7.3 *1.3* | 210.0 ± 7.9 *2.0* | 111.1 ± 4.0 *1.2* | - | - |
| Proline | 210.2 ± 7.7 *3.9* | 604.4 ± 20.8 *4.0* | 614.4 ± 23.1 *5.8* | 496.2 ± 18.5 *5.3* | 38.2 ± 1.4 *9.8* | 15.9 ± 0.6 *2.3* |
| Glycine | 389.7 ± 14.7 *7.33* | 1190.4 ± 44.8 *7.9* | 801.6 ± 30.1 *7.6* | 599.1 ± 22.5 *6.4* | 19.1 ± 0.7 *4.9* | 53.8 ± 2.0 *7.9* |
| Alanine | 450.7 ± 16.9 *8.5* | 1270 ± 46.2 *8.5* | 1475.9 ± 55.5 *14.0* | 912.6 ± 34.3 *9.7* | 15.3 ± 0.6 *3.9* | 28.4 ± 1.1 *4.2* |
| α-aminobutyric acid | - | 1280.6 ± 45.9 *8.7* | 863.7 ± 32.5 *8.2* | 945.9 ± 35.5 *10.1* | 19.0 ± 0.7 *4.9* | 28.0 ± 1.2 *4.1* |
| Valine | 453.3 ± 17.0 *8.5* | 1046.0 ± 39.2 *6.9* | 718 ± 27.0 *6.8* | 777.5 ± 29.2 *8.3* | 10.3 ± 0.4 *2.7* | 15.8 ± 0.6 *2.3* |
| Cysteine | 233.7 ± 8.8 *4.4* | - | - | - | - | - |

**Table 3.** *Cont.*

| Amino Acids | Isaria farinosa | Isaria tenuipes | Isaria fumosorosea | Lecanicillium lecanii | Conidiobolus coronatus | Isaria coleopterorum |
|---|---|---|---|---|---|---|
| Methionine | 65.9 ± 2.5 *1.2* | - | - | - | - | - |
| Cystatathionine | 175.6 ± 6.6 *3.3* | 465.9 ± 17.5 *3.1* | 352.3 ± 13.2 *3.3* | 346.0 ± 13.0 *3.7* | 5.7 ± 0.2 *1.5* | 13.6 ± 0.5 *1.9* |
| Isoleucine | 457.6 ± 17.2 *8.6* | 1391.7 ± 52.3 *9.3* | 968.7 ± 36.4 *9.2* | 1000.8 ± 35.8 *10.7* | 23.9 ± 0.9 *6.1* | 75.0 ± 2.8 *10.9* |
| Leucine | 30.6 ± 1.2 *0.6* | 52.2 ± 1.9 *0.4* | 54.3 ± 2.0 *0.5* | 18.3 ± 0.7 *0.2* | 5.7 ± 0.2 *1.5* | - |
| Tyrosine | 206.1 ± 7.8 *3.9* | 603.3 ± 22.7 *4.0* | 427.5 ± 16.1 *4.1* | 467.0 ± 17.6 *4.9* | 16.8 ± 0.6 *4.3* | 42.1 ± 1.9 *6.2* |
| β-alanine | 8.3 ± 0.3 *0.2* | - | - | - | - | - |
| β-aminobutyric acid | 35.1 ± 1.3 *0.7* | - | - | - | - | - |
| γ-aminobutyric acid | 516.4 ± 19.4 *9.7* | 53.6 ± 2.0 *0.4* | 38.7 ± 1.5 *0.4* | 41.3 ± 1.5 *0.4* | 3.6 ± 0.1 *0.9* | 0.9 ± 0.03 *0.1* |
| Ornithine | 21.5 ± 0.8 *0.4* | 17.6 ± 0.7 *0.1* | 16.0 ± 0.6 *0.2* | - | - | - |
| Lysine | 194.4 ± 7.3 *3.7* | 937.3 ± 35.2 *6.3* | 485.7 ± 18.3 *4.6* | 412.7 ± 15.5 *4.4* | 14.8 ± 0.6 *3.8* | 30.6 ± 1.2 *4.5* |
| Histidine | 69.7 ± 2.6 *1.3* | 232.8 ± 8.8 *1.6* | 83.9 ± 3.2 *0.8* | 53.0 ± 1.9 *0.5* | 4.1 ± 0.2 *1.1* | 4.1 ± 0.2 *0.6* |
| 1-methylohistidine | - | 447.9 ± 16.8 *3.0* | 150.9 ± 5.6 *1.4* | 173.4 ± 6.5 *1.9* | 9.0 ± 0.3 *2.3* | 25.7 ± 1.0 *3.9* |
| 3-methylohistidine | 32.2 ± 1.2 *0.6* | 587.2 ± 22.1 *3.9* | 407 ± 15.3 *3.9* | 318.0 ± 11.0 *3.4* | 7.9 ± 0.3 *2.0* | 28.5 ± 1.1 *4.2* |
| Arginine | 121.7 ± 4.6 *2.3* | 1762.4 ± 66.2 *11.8* | 1129.3 ± 42.3 *10.7* | 1095.6 ± 41.2 *11.7* | 53.6 ± 2.0 *13.7* | 127.1 ± 4.8 *18.6* |
| Total amount | 5317.9 | 14,963.0 | 10,530.41 | 9390.51 | 390.0 | 683.9 |



**Table 4.** Concentration of sulfuric, acidic, neutral and basic amino acids in fungi in µg g$^{-1}$d.m. and %.

| Fungus Name | Amino Acids | | | | | | | |
|---|---|---|---|---|---|---|---|---|
| | Sulfuric | | Acidic | | Neutral | | Basic | |
| | (µg g$^{-1}$d.m.) | (%) | (µg g$^{-1}$d.m.) | (%) | (µg g$^{-1}$d.m.) | (%) | (µg g$^{-1}$d.m.) | (%) |
| *Isaria farinose* | 510.6 | 4.9 | 1781.3 | 16.9 | 5963.5 | 56.6 | 2271.4 | 21.6 |
| *Isaria tenuipes* | 410.5 | 4.4 | 1667.6 | 17.8 | 5259.6 | 56.0 | 2052.8 | 21.9 |
| *Isaria fumosorosea* | 639.5 | 4.3 | 2845.3 | 19.0 | 7492.7 | 50.1 | 3985.2 | 26.6 |
| *Lecanicillium lecanii* | 32.2 | 8.3 | 116.6 | 29.9 | 151.9 | 38.9 | 89.4 | 22.9 |
| *Conidiobolus coronatus* | 32.7 | 4.8 | 174.4 | 25.5 | 259.7 | 37.9 | 216.9 | 31.7 |
| *Isaria coleopterorum* | 571.4 | 10.8 | 1255.5 | 23.6 | 3051.5 | 57.4 | 439.5 | 8.3 |

**Table 5.** Concentration of hydrophilic and hydrophobic amino acids in fungi in µg g$^{-1}$d.m. and %.

| Fungus Name | Amino Acids | | | |
|---|---|---|---|---|
| | Hydrophilic | | Hydrophobic | |
| | (µg g$^{-1}$s.m.) | (%) | (µg g$^{-1}$s.m.) | (%) |
| *Isaria farinose* | 4214.0 | 40.0 | 6315.8 | 59.9 |
| *Isaria tenuipes* | 3784.9 | 40.3 | 5605.6 | 59.7 |
| *Isaria fumosorosea* | 7004.3 | 46.8 | 7958.7 | 53.2 |
| *Lecanicillium lecanii* | 232.3 | 59.6 | 157.6 | 40.4 |
| *Conidiobolus coronatus* | 410.4 | 60.0 | 273.3 | 39.7 |
| *Isaria coleopterorum* | 1791.2 | 33.68 | 3526.7 | 66.32 |

In all analyzed fungi samples, aspartic acid, hydroksyproline, glycine, alanine, isoleucine and arginine dominated (Table 3). In the studied entomopathogenic fungi, except *Isaria farinosa*, the highest concentration measured for arginine was from 11 to 19%. Its highest content among all identified amino acids was found in *Isaria coleopterorum*, while the lowest was in *Isaria fumosorosea*. Arginine is one of a number of isoenzymes presenting in the fungal cell walls [8].

In the mycelium of *Isaria farinosa* hydroxyproline dominated. Its content was about 20% of all amino acids determined. This amino acid is formed from proline as a result of metabolic pathways in the mycelium [8].

Among neutral amino acids, the highest content for isoleucine and glycine was determined. The concentration of isoleucine in mycelium ranged from 6 to 11% and its highest concentration was demonstrated in *Isaria coleopterorum*. The percentage of glycine in the tested fungi ranged from 5 to 8% and its highest concentrations were found in *Isaria tenuipes*, while the lowest contents, as in the case of isoleucine, were reported for *Isaria coleopterorum*.

Aspartic acid dominated among the acidic amino acids except for *Isaria farinosa*. The highest percentage of its content was found in the mycelium of *Isaria coleopterorum* (6.7% of total amino acids).

Analyzing the elements characterizing the pathogenicity of individual species of fungi based on infectivity criteria, ranges of infected hosts and the ability to induce epizootics, these can be ranked in the order shown in Table 3. The high infectivity of the strain *Isaria tenuipes* (*Paecilomyces sp.II*) has been identified only in experimental conditions, but there are no studies on its pathogenicity in nature [33]. Therefore, the richest and the largest amounts of amino acids are formed by strains of the species demonstrating the strongest entomopathogenic properties (Table 3).

**Table 6.** Content of bonded amino acids in entomopathogenic fungi determined in our study and by other authors.

| Amino Acids | *Isaria farinose* | | *Isaria tenuipes* | | *Isaria fumosorosea* | | *Lecanicillium lecanii* | | *Conidiobolus coronatus* | | *Isaria coleopterorum* |
|---|---|---|---|---|---|---|---|---|---|---|---|
| | Our Study | [51] | Our Study | [52] | Our Study | [53] | Our Study | [49] | Our Study | [54] | Our Study |
| Alanine | + | + | + | + | + | | + | + | + | + | + |
| β-alanine | + | | | | | | | | | | |
| α-aminobutyric acid | | | + | | + | | + | | + | | + |
| β-aminobutyric acid | + | | | | + | | | | | | |
| γ-aminobutyric acid | + | | + | | + | | + | | + | | + |
| Arginine | + | | + | + | + | | + | | + | + | + |
| Asparagine | | + | | | | + | | | | + | |
| Aspartic acid | + | | + | + | + | | + | | + | | + |
| Cystathionine | + | | + | | + | | + | | + | | + |
| Cysteic acid | + | | + | | + | | + | | + | | + |
| Cysteine | + | + | | | | | | | | | |
| Glutamic acid | + | + | + | + | + | | + | + | | | |
| Glutamine | | | | | | | | | | + | |
| Glycine | + | + | + | + | + | | + | | + | + | + |
| Histidine | + | + | + | + | + | + | + | | + | + | + |

**Table 6.** *Cont.*

| Amino Acids | *Isaria farinose* | | *Isaria tenuipes* | | *Isaria fumosorosea* | | *Lecanicillium lecanii* | | *Conidiobolus coronatus* | | *Isaria coleopterorum* |
|---|---|---|---|---|---|---|---|---|---|---|---|
| | Our Study | [51] | Our Study | [52] | Our Study | [53] | Our Study | [49] | Our Study | [54] | Our Study |
| Hydroksyproline | + | | + | | + | | + | | + | | + |
| Isoleucine | + | + | + | + | + | | + | | + | + | + |
| Leucine | + | + | + | + | + | | + | | + | + | |
| Lysine | + | + | + | + | + | | + | + | + | + | + |
| Methionine | + | + | | + | | | | | | + | |
| 1-methylohistidine | | | + | | + | | + | | + | | + |
| 3-methylohistidine | + | | + | | + | | + | | + | | + |
| Ornithine | + | | + | | + | | | | | | |
| Phenylalanine | | + | | + | | | | | | + | |
| Phosphoethanolamine | | | | | | | | | + | | + |
| Proline | + | + | + | + | + | | + | + | + | + | + |
| Serine | + | + | + | + | + | + | + | | + | + | + |
| Taurine | + | | + | | + | | + | | + | | + |
| Threonine | | + | + | + | + | | + | | + | + | + |
| Tryptophan | | | | | | | | | | + | |
| Tyrosine | + | | + | + | + | | + | + | + | + | + |
| Valine | + | + | + | + | + | | + | | + | + | + |

+ Positively determined.

The largest number of amino acids was isolated from the mycelium strain of *Isaria farinosa* (*Paecilomyces farinosus*), wherein the four compounds: cysteine, methionine, β-alanine and β-aminobutyric acid do not occur in any of the other tested strains. This may suggest that their presence in the composition of the fungus metabolites affects the scope of its affinity for the host pathogenic because both the number and taxonomic differentiation of arthropods infected by *Isaria farinosa* (*Paecilomyces farinosus*) are the greatest of all the tested strains.

Special attention is given to the very high content of hydroxyproline in the mycelium strain of *Isaria farinosa* in comparison with the other species (*Isaria tenuipes, Isaria fumosorose, Lecanicillium lecanii, Conidiobolus coronatus, Isaria coleopterorum*). This amino acid is created from proline as a result of metabolic conversions in the mycelium.

However, α-aminobutyric acid, threonine, and 1-methylohistidine not produced by the *Isaria farinosa* (*Paecilomyces farinosus*) strain occurred in all other species. α-aminobutyric acid belongs to dominants in three strains of high pathogenicity. Another characteristic feature is the similar and proportionally high share of five amino acids (glycine, alanine, valine, isoleucine, leucine) in hydrolysates of the four most pathogenic fungal strains [31,33].

The strains of *Conidiobolus coronatus, Isaria coleopterorum* are slightly different from the rest of the composition and the percentage share of the produced amino acids. However, the absolute amount of these compounds in hydrolysates of mycelia was over 10 to 40 times lower than in highly virulent species. In comparison with *Isaria farinosa* amino acids, the composition of the mycelium *Conidiobolus coronatus* was poorer for six of the compounds (25%) and seven compounds (29%) relative to that of the mycelium of *Isaria coleopterorum*. The last two differ from all of the highly pathogenic phosphoethanolamines and lack of production of glutamic acid; in *Isaria coleopterorum* the presence of leucine was not observed. In addition, there was a lack of ornithine in the mycelia hydrolysates of both strains of weak pathogenicity and also in *Lecanicillium lecanii*, which is considered a conditional pathogen.

Growth in the hemocoel includes various hydrolytic enzymes and may lead to the creation of smaller compounds with harmful effects on the host [35]. They are simple organic poisons, e.g., oxalic acid, but these are mainly larger compounds, e.g., depsipeptides. These substances such as metabolites of entomopathogenic fungi are investigated as potential chemical insecticides and as pharmacological agents.

Our preliminary study has shown the relationship between the content of bounded amino acids and the pathogenicity of enthomopathegenic fungi. The pathogenicity of entomopathogenic fungus includes the wide complex of chemical substances—mycotoxins (Table 7). The $IC_{50}$—the half maximal inhibitory concentration, and the $LD_{50}$—lethal dose (50%) shows the effectiveness of these mycotoxins. Therefore, we decided to include the compendium of physicochemical properties and physiological activity of mycotoxins created by entomopathogenic fungi in our paper. The data compendium and our results of bounded amino acids should give a better insight into the chemistry of entomopathogenic fungus.

**Table 7.** Mycotoxins of entomopathogenic fungi, physicochemical properties and physiological activity.

| Fungi | Compounds Molecular Formula (MF), Molecular Weight (MW) | Physiological Activity $IC_{50}$—The Half Maximal Inhibitory Concentration, $LD_{50}$—Lethal Dose, 50% |
|---|---|---|
| 1. *Isaria farinosa* | Farinomalein MF $C_{10}H_{13}NO_4$, MW 211.21 g mol$^{-1}$ | The first natural maleimide, pesticide of Phytophthora stem rot in soybean [55], activity comparable to antibiotic amphotericin B for inhibition *Phytophthora sojae* [56], selective inhibition of *Phytophthora sojae* with an MIC (Minimal Inhibitory Concentration) value of 5 µg disk$^{-1}$ [18], $IC_{50}$ = 4.4 µg mL$^{-1}$ [55,57] |
| | Farinosones A, B, C A-MF $C_{25}H_{27}NO_4$, MW 405.49 g mol$^{-1}$ B-MF $C_{25}H_{26}NO_5$, MW 420.48 g mol$^{-1}$ C-MF $C_{19}H_{25}NO_5$, MW 347.41 g mol$^{-1}$ | Classe as polyketide, cytotoxic in the PC-12 cell line [20], pyridone alkaloid, neurotrophic activities [58], antimicrobial, antitumor, $IC_{50}$ = 1.1 µg mL$^{-1}$ [59] |
| | Paecilosetin MF $C_{22}H_{31}NO_4$, MW 373.49 g mol$^{-1}$ | Tetramic acid derivative, antibacterial, ant-leucemia (P388 cell line) [59] antimicrobial, antitumor, $IC_{50}$ = 3.2 µg mL$^{-1}$ [60] |
| | Leucinostatin A MF $C_{62}H_{111}N_{11}O_{13}$, MW 1218.61 g mol$^{-1}$ | Linear peptides, paecilotoxins, $LD_{50}$ = 1.8 mg kg$^{-1}$-mice, antimicrobial, anticancer [35], antibiotic [61] |
| 2. *Isaria tenuipes* | Extract | $LD_{50}$ = >2000 mg/kg of body weight, $LD_{50}$ = >5 g kg$^{-1}$ of body weight in rats and dogs [62] |
| | Tenuipyrone MF $C_{15}H_{16}O_6$, MW 292.28 g mol$^{-1}$ | Polyketide with an unprecedented tetracyclic ring system bearing a spiroketal structural component [20,63] |
| | Acetoxyscirpenediol MF $C_{17}H_{24}O_6$, MW 324.37 g mol$^{-1}$ | Trichothecenes, cytotoxic, antitumor [59], metabolite of anguidine is to inhibit protein synthesis in rabbit reticulocytes, anguidine is antitumor agent which was in the process of phase II evaluation in gastrointestinal malignancy, central nervous system tumors, colorectal adenocarcinoma [64]) |
| | Spirotenuipesine A, B A-MF $C_{15}H_{22}O_4$, MW 266.34 g mol$^{-1}$ B-MF $C_{15}H_{22}O_5$, MW 282.33 g mol$^{-1}$ | Trichothecenes, neurotrophic factor biosynthesis [59] |

**Table 7.** *Cont.*

| Fungi | Compounds Molecular Formula (MF), Molecular Weight (MW) | Physiological Activity IC$_{50}$—The Half Maximal Inhibitory Concentration, LD$_{50}$—Lethal Dose, 50% |
|---|---|---|
| 3. *Isaria fumosorosea* | Extract (beauverolide 1, beauverolide 5, 2.6-pyridinedicarboxylic) | LD$_{50}$ > 0.9 × 10$^8$ CFU/animal-pulmonary, LD$_{50}$ > 10$^8$ CFU/animal-oral [65] |
| | Fumosorinone MF C$_{29}$H$_{35}$NO$_5$, MW 477.59 g mol$^{-1}$ | Classe as polyketide, inhibits tyrosine phosphatase 1B (PTP1B) to treat type 2 diabetes mellitus (T2DM) [20], 2-pyridone alkaloid, insecticidal, IC$_{50}$ = 14.04 IM [66] |
| | Beauvericin MF C$_{45}$H$_{57}$N$_3$O$_9$, MW 783.95 g mol$^{-1}$ | Cytotoxic, cyclic peptide, insecticidal against mosquito larvae and blowflies, antiplasmodial [35,59], antibiotic, cyclohexadepsipeptide, anthermintic, antibacterial, antifungal, antimycobacterial and anticancer activities [39], herbicidal, antiretroviral, cytotoxic, antihaptotactic, anti-cholesterol, chemosensitizer, as well as repression of amyloid plaque formation in Alzheimer's disease [67], LD$_{50}$ = 100 mg kg$^{-1}$-mice [68], IC$_{50}$ = 0.59 μM [69], LC$_{50}$ of 633 ppm (95% fiducial limits, 530–748 ppm) and LC$_{90}$ of 1196 ppm (95% fiducial limits, 954–1863 ppm) [35] |
| | Beauverolide MF C$_{29}$H$_{45}$N$_3$O$_5$, MW 515.69 g mol$^{-1}$ | Cyclic tetradepsipeptide [59], inhibits the adhesion, extension, and phagocytosis of plasmatocyte of *Galleria mellonella* [70] |
| | Pyridine-2,6-dicarboxylic acid (dipicolinic acid) MF C$_7$H$_5$NO$_4$, MW 167.19 g mol$^{-1}$ | Insecticidal [71], produces pharmaceuticals and metal salts for the application of nutritional supplements, act a chelating agent and an enzyme inhibitor [72], LD$_{50}$ = 20 μg mice$^{-1}$ intramuscularly on the day of infection, LD$_{50}$ = 10.50 g kg$^{-1}$ of body weight [73] |
| | Cepharosporolides C, E, F E-MF C$_{10}$H$_{14}$O$_4$, MW 198.22 g mol$^{-1}$ | Biosynthetic precursor of tenuipyrone, and cephalosporolide F [63,69] |
| | 2-carboxymethyl-4-(30-hydroxybutyl)furan | Insecticidal [74], biological and pharmacological activity, chemotherapeutic agents, IC$_{50}$ = >1000 (μM) [69,75] |
| | Enniatins A, B, C (fusafungine) A, C-MF C$_{36}$H$_{63}$N$_3$O$_9$, MW 681.90 g mol$^{-1}$ B-MF C$_{34}$H$_{59}$N$_3$O$_9$, MW 653.85 g mol$^{-1}$ | Microbial, a cyclodepsipeptide alkali metal ionophore inhibitor of acyl-CoA: cholesterol acyltransferase (ACAT), antitumor (Sigma), antibiotic activity, toxins in low non-toxic concentrations are tested for application in human and veterinary medicine, phytotoxic, herbicide, IC$_{50}$ = 10$^{-4}$ M effect on seed germination of the parasitic weed *Striga hermonthica* [35,76], LD$_{50}$ = 350 mg kg$^{-1}$-mice [68] |

Table 7. *Cont.*

| Fungi | Compounds Molecular Formula (MF), Molecular Weight (MW) | Physiological Activity IC$_{50}$—The Half Maximal Inhibitory Concentration, LD$_{50}$—Lethal Dose, 50% |
|---|---|---|
| 4. *Lecanicillium lecanii* | Bassianolide MF C$_{48}$H$_{84}$N$_4$O$_{12}$, MW 909.20 g mol$^{-1}$ | Classe as nonribosomal peptide, insecticidal [20], toxic to lepidopteran larvae after infection or feeding [77], biological activities for cryptophycins, didemnins, dolastatins, PF1022, enniatin, destruxin, beauvericin, valinomycin and anti-plasmodial, antimycobacterial, antitumor activities, toxicity against bacteria: *Staphylococcus aureus, Bacillus subtilis, Escherichia coli, Pseudomonas aeruginosa* [78] |
| | Cyclosporine MF C$_{62}$H$_{111}$N$_{11}$O$_{12}$, MW 1202.61 g mol$^{-1}$ | Cyclic peptide, immunosuppressant, insecticidal (mosquito, but no *Galleria mellonella*) [38,59,79], restaining rejection following solid organ transplantations especially heart, lung, kidney, preventing and treating graft-versus-host disease after bone marrow transplants, the treatment of numerous autoimmune diseases, anti-inflammatory, anti-parasitic (anti-malaria), antifungal, antiviral (ant-HIV) [79], LD$_{50}$ = 2329 mg kg$^{-1}$-mice, 1480 mg kg$^{-1}$-rats, >1000 mg kg$^{-1}$-rabbits (DrugBank online) |
| | Pyridine-2,6-dicarboxylic acid (dipicolinic acid) | Look at *Isaria fumosorosea* [59,71] |
| | Hydroxycarboxilic acid | Group of chemicals use in processes such as biodegradable plastics for consumer products and nontoxic and easily degradable solvents, cleaning agents, plasticizers, etc., global commercial [59,80] |
| | Verlamelin A,B B-MF C$_{44}$H$_{69}$N$_7$O$_{11}$, MW 872.06 g mol$^{-1}$ | Cyclic hexadepsipeptide lipopeptide, exhibits antifungal activity against plant pathogenic fungi [81], drugs [81,82] |
| | Vertilecanin A, B, C A-MF C$_{13}$H$_{11}$NO$_3$, MW 229.21 g mol$^{-1}$ | Phenopicolinic acid derivatives were synthesized from nicotinic acid, antibacterial activity against *Bacillus subtilis* [83,84], insecticidal activity against *Helicoverpa zea* [85] |
| | Phospholipids toxic | Insecticidal [86] |

<div align="center">**Table 7.** *Cont.*</div>

| Fungi | Compounds Molecular Formula (MF), Molecular Weight (MW) | Physiological Activity IC$_{50}$—The Half Maximal Inhibitory Concentration, LD$_{50}$—Lethal Dose, 50% |
|---|---|---|
| 5. *Conidiobolus coronatus* | Fumonisin B1 MF $C_{34}H_{59}NO_{15}$, MW 721.83 g mol$^{-1}$ | Bioinsecticide [39] causes leukoencephalomalacia in horses, mycotoxin, hepatotoxic (Sigma), nephrotoxicity, immunotoxicity, reproductive toxicity, embryotoxicity, teratogenicity, mutagenicity, carcinogenicity, phytotoxic, nephrotoxic, pulmonary, oedema syndrome, hepatocarcinogenic, esophageal cancer [87], LD$_{50}$ = 1.25 mg kg$^{-1}$ [88] |
| | Destruxin A MF $C_{29}H_{47}N_5O_7$, MW 577.71 g mol$^{-1}$ | Cyclic depsipeptides consisting of 5 amino acids and D-α-hydroxy acid, insecticidal [35], antitumor, herbicidal, cytoxicity [20], bioinsecticide [39,89], phytotoxin, cancer cell, anti-immunity of *Metarhizium anisopliae* antimicrobial peptides [89], virustatic against, LD$_{50}$ = 13.2–16.9 mg/kg-mice [3] |
| | Enniatin A, B (fusafungine) A-MF $C_{36}H_{63}N_3O_9$, MW 681.90 g mol$^{-1}$ B-MF $C_{33}H_{57}N_3O_9$, MW 639.00 g mol$^{-1}$ | Look at *Isaria fumosorosea* |
| | Beauvericin | Look at *Isaria fumosorosea* |
| | T-2 and HT-2 toxins metabolites from trichothecene group T-2-MF $C_{24}H_{34}O_9$, MW 466.52 g mol$^{-1}$ HT-2-MF $C_{22}H_{32}O_8$, MW 424.49 g mol$^{-1}$ trichothecene MF $C_{15}H_{22}O_2$, MW 234.16 g mol$^{-1}$ | Bioinsecticide [39], mycotoxin, antibacterial, antiviral, antifungal, cytostatic activity, anorexia, hematuria, leukocytosis, leukopenia, inhibitors of protein and DNA synthesis, natural intoxications of animals and man [36], sesquiterpenoid chemicals characterized by a tetracyclic 12,13-epoxytrichothecene, LD$_{50}$ = 0.5–300 mg kg$^{-1}$-animal, LD$_{50}$ = 13–140 μg kg$^{-1}$-rat brain [90] |
| 6. *Isaria coleopterorum* | | Pathogens of *Lepidoptera* and *Coleoptera* [91] |

Table 7 shows the list of mycotoxins produced by examined entomopathogenic fungi in our study and their physicochemical properties, and physiological activity. These substances represent a heterogeneous group of metabolites that are formed along several biochemical pathways [8,36–39]. The above mycotoxins are produced from a few key intermediates of primary metabolism, e.g., acetate, propionate, pyruvate, malonate, mevalonate, and amino acids by a subsequent series of enzyme-catalyzed reactions. In view of their structural complexity and unique physiological properties, these highly specialized secondary microbial metabolites, which are characteristic of the lower forms of life, are species-specific. Some of the poisons recently extracted and isolated from these fungi are chemically similar to drugs well known from medicinal plants. These fungi have been found to contain biologically active substances belonging to certain categories of poisons. For example, Beauvericin produced by *Isaria fumosorosea*, Bassianolode and Verlamerin by *Lecanicillium lecanii* and Destruxin A by *Cinidiobolus coronatus* are well known toxin agents with low values of IC 50 (Table 7). These compounds may be classed as truly poisonous in the conventional sense of the term and may be active in relatively low concentrations. However, a hypothesis is still unclear and should be explain how the fungal cells might produce poisons without poisoning themselves. The hypothesis takes into account basic characteristic of the formation of biological structures, such as fungus cell membranes.

On the basis of the obtained results, it cannot be concluded whether the low content of amino acids in the mycelium of *Conidiobolus coronatus* and *Isaria coleopterorum* is the result of their poor pathogenicity or the use of too poor a composition of the culture medium. The latter hypothesis seems more probable. Czapek's mineral medium is widely used primarily as a standard culture medium for the saprophytic *Hyphomycetes* and *Zygomycetes*, especially *Mucorales* [40]. However, it is not considered optimal for pathogenic fungi.

For fungus, some authors postulated a varied composition of medium which modifies the quantity and quality of metabolites and mycotoxins. Therefore, the composition of the medium may be an important factor in regulating the pathogenicity of the fungus.

For example, Koval [41] recommends enriching it with insect decoction. This medium, however, is often used in various modifications due to the strictly defined chemical composition, allowing to observe a number of metabolites. For this reason, it was also used in these studies.

In addition, the development of *Conidiobolus coronatus* and *Isaria coleopterorum* on media with various composition shows that their nutritional requirements are greater than the four strains showing high pathogenicity. The study results on the physiology of highly specialized entomopathogenic fungi, especially *Entomophthorales* and *Ascomycetes* of *Cordyceps* genus and related ones including theirs anamorphs belonging to the group of *Hyphomycetales*, indicate that in most cases, these fungi require for development an in vitro of the substrates enriched with various organic compounds for their development, such as, e.g., fatty acids, vitamins, meat extracts and egg yolks. With these substances, numerous amino acids are also introduced into the media among other chemical compounds. Their impact on developing cultures and the degree of use as sources of nutrients are difficult to clearly determine [42].

Nolan [43] demonstrated, as a result of a long-term study into the development of protoplasts of various *Entomophaga aulicae* (Reihardt in Bail) Batko strains, that they are autotrophic in vitro to all organic acids and vitamins. From among 20 amino acids added to the medium, whose composition was similar to those discussed in the present study, none was the only source of nitrogen. Most were used in a non-uniform manner by various strains of fungi, for example, in a medium that does not contain any absorption of serine, glutamine, alanine, β-alanine by fungi in its composition of the calf germ serum. The results of the presented studies have simplified the composition of amino acids in the medium used for the cultivation of *Entomophaga aulicae* protoplasts from 21 in the Grace recipe based medium [44] (with the addition of calf germ serum) to 8 in the final medium enriched with hematine and oleic acid.

Several authors suggested that the composition of the medium may have a significant influence on the pathogenic properties of the strains because they are related to the biological activity of enzymes,

formation of toxins, phytohormones and pathogen antibodies. In addition, the physicochemical, physiological properties of amino acids as components of peptide chains may be indirect factors characterizing the pathogenic potential of individual entomopathogenic fungal strains [45–50]. Due to the high diversity of qualitative and functional protein components in entomopathogenic fungi, it is important to pay attention to the composition and function of amino acids in the individual development of these pathogens and the disease processes caused by them.

## 4. Conclusions

A high-resolution column packed with Ostion LGFA of Amino Acid Analyzer and the system of lithium-cytric buffers is recommended for the separation and determination of the larger number of amino acids −24 in mycelium hydrolysates.

Our preliminary study indicates that there are significant quantitative and qualitative differences of amino acids in the entomopathogenic fungal strains contained in the mycelium between high and low pathogenicity strains. Analyzing the elements characterizing the pathogenicity of individual species of fungi based on infectivity criteria, ranges of infected hosts, and the ability to induce epizootics, these can be ranked in the following order: *Isaria farinosa, Isaria tenuipes, Isaria fumosorose, Lecanicillium lecanii, Conidiobolus coronatus, Isaria coleopterorum.*

The richest composition of amino acids has been found in the mycelium of *Isaria farinosa* strain, which most commonly exhibits pathogens in this group of fungi. A wide range of infected hosts and their high frequency recorded in nature are cited as important factors limiting the population of insects. In the mycelium strain of *Isaria farinosa* (*Paecilomyces farinosus*), the four compounds: cysteine, methionine, β-alanine and β-aminobutyric acid do not occur in any of the other tested strains. This may suggest that their presence in the composition of the fungus metabolites affects the scope of its affinity for the host pathogenic response because both the number and taxonomic differentiation of arthropods infected by *Isaria farinosa* (*Paecilomyces farinosus*) are the greatest of all the tested strains.

In addition, in the mycelium strain of *Isaria farinose* a very high content of hydroxyproline in comparison with the other species was determined. This amino acid is formed from proline as a result of metabolic pathways in the mycelium.

α-aminobutyric acid, threonine, and 1-methylohistidine not produced by the *Isaria farinosa* (*Paecilomyces farinosus*) strain occurred in all other species. α-aminobutyric acid belongs to dominants in three strains of high pathogenicity. Another characteristic feature is the similar and proportionally high share of five amino acids (glycine, alanine, valine, isoleucine, leucine) in hydrolysates of the four most pathogenic fungal strains.

In *Isaria farinosa, Isaria tenuipes, Isaria fumosorosea, Leanicillium lecanii*, a group of hydrophobic amino acids dominated and their content ranged from 53 to 66%, while hydrophilic amino acids ranged from 34 to 47% in the tested entomapathogenic fungi. With reference to *Conidiobolus coronatus* and *Isaria coleopterorum*, the dominant groups in the hydrolysates from their mycelium were generally hydrophilic amino acids, whose content was close to approximately 60% and hydrophobic amino acids at 40%, respectively.

Aspartic acid, hydroksyproline, glycine, alanine, isoleucine, and arginine dominated in all analyzed fungi samples. In the studied entomopathogenic fungi, except *Isaria farinosa*, the highest concentration of arginine was from 11 to 19%. Its highest content among all identified amino acids was found in *Isaria coleopterorum*, while the lowest was in *Isaria fumosorosea*. Arginine is involved in several isoenzymes found in the fungal cell walls.

The results obtained from our study should give a better insight into the content of bounded amino acids in the entomopathogenic fungi and should be helpful in more quantitative characterization of the pathogenicity of this group of fungi.

**Author Contributions:** Conceptualization, L.W.S., S.B.; methodology L.W.S., S.B., software, T.M.; validation, L.W.S.; formal analysis, L.W.S., T.M.; investigation, L.W.S., T.M.; resources, S.B.; data curation, L.W.S., T.M.; writing—original draft preparation, L.W.S.; writing—review and editing, L.W.S., S.B.; supervision, L.W.S. All authors have read and agreed to the published version of the manuscript.

**Funding:** This research received no external funding.

**Conflicts of Interest:** The authors declare no conflict of interest.

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
