# Peer review of "Amino Acids in Entomopathogenic Fungi Cultured In Vitro"

_agronomy, doi:10.3390/agronomy10121899_

Round 1
Reviewer 1 Report
The paper entitled “Amino acids in entomopathogenic fungi cultured in vitro” is a very interesting study.
- Some background information should be included in the abstract section before presenting the goal of the study.
- The introduction should be shorter and better explained. For example, the relevance of determining amino acids should be emphasized more rather than listing the different works that have studied them.
- The material and methods section details very well the information about the different species. However, there is very little information on the analytical methodology used to determine amino acids. Please include more information about the technique, method, standards used, etc.
- A quantification has been performed so the methodology should be validated. Please, include information about the quantification procedure (calibration curves, standards, accuracy, precision, linearity, etc.)
- The paragraph “Where: 1-cysteic acid, 2-taurine, …” should be written together with the figure caption of the Figure 2
- Line numbers and agronomy template should have used.
- Correct Significant numbers/figures in the confidence intervals.
- Some sections of the introduction and discussion of results seem more typical of a review.
- The authors should focus more on discussing their results as well as interpreting the biological role of the amino acids detected, rather than describing the rest of the studies.
Author Response
Dear Reviewer No1. Thank you for your review. All your suggestions, proposals were pertinent. They all significantly increase the scientific value of the paper.
Below you will see the answers for your each point (bold marked).
All the changes, modification in the text are red marked.
Thank you for your help. The new version presents a higher scientific level than the previous one.
My best regards
Lech W. Szajdak
- Some background information should be included in the abstract section before presenting the goal of the study.
The abstract is new. There is the goal of the study in the end of introduction.
- The introduction should be shorter and better explained. For example, the relevance of determining amino acids should be emphasized more rather than listing the different works that have studied them.
The introduction is changed according to you suggestion.
- The material and methods section details very well the information about the different species. However, there is very little information on the analytical methodology used to determine amino acids. Please include more information about the technique, method, standards used, etc.
The section: material and method is new. I added many details about this method for the determination and separation of bounded amino acids. You will find information about the content of standards, parameters and properties of columns, system of buffers, properties of ninhydrin.
I have published many papers on the determination of biologically active compounds in soils with various analytical methods HPLC, GS, fluorimetric, UV-Vis spectroscopy etc. However, the reviewers were against the listing of exact methods in the methods section. You are the first to ask me for it. It was a great pleasure for me to be able to present all the details.
Reviewer 2 Report
I have reviewed the manuscript titled “Amino acids in entomopathogenic fungi cultured in vitro” by Szajdak et al.
The authors set out to quantity and qualify the different amino acids associated with entopathogenic fungi and promised this discuss their results in relation to the pathogenicity of the different strains. Unfortunately this discussion was rather weak possibly because, sometimes, the grammar was hard to understand. Secondly, there was often no logical flow of information.
Please find below specific consideration in the different sections
Introduction: Important background information highlighting the significance of this study and how it contributes to the advancement of science was noticeably absent.
Literature review: The argument in this section was particularly weak as it was not backed by sufficient up to date literature in the subject area.
Methodology: The methodology did not provide sufficient details to replicate the study
Results and Discussion: Some sections of the results and discussion sounded like literature review as the authors failed to relate this information to their results.
Details on specific comments have been highlighted in the manuscript.

Author Response
Dear Reviewer No2. Thank you for your review. All your suggestions, proposals were pertinent. They all significantly increase the scientific value of the paper.
Below you will see the answers for your each point (bold marked).
All the changes, modification in the text are red marked.
Thank you for your help. The new version presents a higher scientific level than the previous one.
My best regards
Lech W. Szajdak
The abstract, introduction, and conclusion have been rewritten. The abstract is new. There is the goal of the study in the end of introduction.
The section: material and method is new. I added many details about this method for the determination and separation of bounded amino acids. You will find information about the content of standards, parameters and properties of columns, system of buffers, properties of ninhydrin.
I would like to keep the text which combines the results of my research with the results of other authors. I think it may be interesting for the reader and improve the quality of the paper.
The authors set out to quantity and qualify the different amino acids associated with entopathogenic fungi and promised this discuss their results in relation to the pathogenicity of the different strains. Unfortunately this discussion was rather weak possibly because, sometimes, the grammar was hard to understand. Secondly, there was often no logical flow of information.
Please find below specific consideration in the different sections
Introduction: Important background information highlighting the significance of this study and how it contributes to the advancement of science was noticeably absent.
Literature review: The argument in this section was particularly weak as it was not backed by sufficient up to date literature in the subject area.
The literature review has been rewritten.
Methodology: The methodology did not provide sufficient details to replicate the study
The section: material and method is new. I added many details about this method for the determination and separation of bounded amino acids. You will find information about the content of standards, parameters and properties of columns, system of buffers, properties of ninhydrin.
Results and Discussion: Some sections of the results and discussion sounded like literature review as the authors failed to relate this information to their results.
I would like to keep the text which combines the results of my research with the results of other authors. I think it may be interesting for the reader.
Our preliminary study has shown the relationship between the content of bounded amino acids and the pathogenicity of enthomopathegenic fungus. The pathogenicity of entomopathogenic fungus includes the wide complex of chemical substances – mycotoxins (Table 7). The IC50 - the half maximal inhibitory concentration, and the LD50 - lethal dose, 50% shows the effectiveness of these mycotoxins. Therefore we decided to include in our paper the compendium of physicochemical properties and physiological activity of mycotoxins created by entomopathogenic fungus . The data compendium and our results of bounded amino acids should give a better insight into the chemistry of entomopathogenic fungi.
. Details on specific comments have been highlighted in the manuscript.
Round 2
Reviewer 1 Report
The new version has been improved according to reviewers' comments. However, I consider that the abstract has not been improved correctly. Now, its too long and difficult to read. Please rewrite this part making it shorter and focusing on commenting on the background, the reasons for carrying out the study, present the objective and highlighting the most relevant studies. Do not include methodological information. The abstract is the most important section so that other researchers are interested in the article. Please correct this carefully.
Author Response
Dear Reviewer No 1.
Thank you for your review.
The abstract was shortened and rewritten.
Below I present a new version of the abstract (red marked)
Thank you for your help. The new version of abstract presents a higher scientific level than the previous one.
My best regards
Lech W. Szajdak
The content of bounded amino acids in the six entomopathogenic fungi was identified and determined. Analyzing the elements characterizing the pathogenicity of individual species of fungi based on infectivity criteria, ranges of infected hosts, and the ability to induce epizootics, these can be ranked in the following order: Isaria farinosa, Isaria tenuipes, Isaria fumosorose, Lecanicillium lecanii, Conidiobolus coronatus, Isaria coleopterorum. These fungi represent two types of Hyphomycetales-Paecilomyces Bainier and Verticillium Nees ex Fr. and one type of Entomophtorales-Conidiobolus Brefeld.
Our study indicates that there are significant quantitative and qualitative differences of bounded amino acids in the entomopathogenic fungal strains contained in the mycelium between high and low pathogenicity strains.
The richest composition of bounded amino acids has been shown in the mycelium of the Isaria farinosa strain, which is one of the most commonly presenting pathogenic fungus in this group with a very wide range of infected hosts and the most frequently recorded in nature as an important factor limiting the population of insects.
Reviewer 2 Report
I have reviewed the manuscript titled “Amino acids in entomopathogenic fungi cultured in vitro” by Szajdak et al.
The authors set out to quantity and qualify the different amino acids associated with entomopathogenic fungi and discussed their results in relation to the pathogenicity of the different strains. This manuscript has been greatly improved however; they are still some minor corrections that need to be made addressed before it is published
1. The abstract needs restructuring. It is important to following the journals limits of just 200 words and also ensure that only key information in the different sections are
2. Some parts of the manuscript will benefit from restructuring the paragraphs following guidelines stipulated by the section 3.2 of the journal (https://www.mdpi.com/authors/layout#_bookmark8). It is important to first to introduce the main idea, then give further relevant details, and finally to give interpretations or conclusions. This structure gives more clarity to readers otherwise the reader may get lost trying to figure out what message the authors are trying to convey. As much as possible please avoid those short paragraphs with one or two sentences. A new paragraph should bring in a near idea.
3. The key words section should be after the abstract and not after the methodology section
4. The information on the Characterisation of fungi is very important with extremely beautiful pictures to compliment it. Unfortunately, that information does not fall in the methodology section. I will suggest you move these important details to the supplementary section so the manuscript is short and concise. Remember the goal is to keep your readers interested.
5. Please ensure the tenses used in the methodology are in accordance with section 4.1.2 of the journal’s guidelines which can be found here https://www.mdpi.com/authors/layout#_bookmark21
Comments from the previous draft that were not address
Page 6, Line 213: what kind of sieve was used to separate the particle sizes. This information is important to ensure the work can be repeated
Page 6 line 223 to 226
It is not clear how cultures in petri dishes were washed thrice with dd water
It is not clear what kind of filters were used
It is not clear how long the samples were dried at 80oC
More specific comments can be found on the manuscript

Author Response
Reviewer 2
Dear Reviewer No 2.
Thank you for your review and suggestions. They all significantly increase the scientific value of the paper.
Below you will see the answers for your each point (red marked).
All the changes, modification in the text are red marked.
Thank you for your help. The new version of the manuscript presents a higher scientific level than the previous one.
My best regards
Lech W. Szajdak
1. R The abstract needs restructuring. It is important to following the journals limits of just 200 words and also ensure that only key information in the different sections are
The abstract was shortened and rewritten.
Below I present a new version of the abstract:
The content of bounded amino acids in the six entomopathogenic fungi was identified and determined. Analyzing the elements characterizing the pathogenicity of individual species of fungi based on infectivity criteria, ranges of infected hosts, and the ability to induce epizootics, these can be ranked in the following order: Isaria farinosa, Isaria tenuipes, Isaria fumosorose, Lecanicillium lecanii, Conidiobolus coronatus, Isaria coleopterorum. These fungi represent two types of Hyphomycetales-Paecilomyces Bainier and Verticillium Nees ex Fr. and one type of Entomophtorales-Conidiobolus Brefeld.
Our study indicates that there are significant quantitative and qualitative differences of bounded amino acids in the entomopathogenic fungal strains contained in the mycelium between high and low pathogenicity strains.
The richest composition of bounded amino acids has been shown in the mycelium of the Isaria farinosa strain, which is one of the most commonly presenting pathogenic fungus in this group with a very wide range of infected hosts and the most frequently recorded in nature as an important factor limiting the population of insects.
2. Some parts of the manuscript will benefit from restructuring the paragraphs following guidelines stipulated by the section 3.2 of the journal (https://www.mdpi.com/authors/layout#_bookmark8). It is important to first to introduce the main idea, then give further relevant details, and finally to give interpretations or conclusions. This structure gives more clarity to readers otherwise the reader may get lost trying to figure out what message the authors are trying to convey. As much as possible please avoid those short paragraphs with one or two sentences. A new paragraph should bring in a near idea.
Thank you for your suggestion. The parts of the text have been shortened and changed according to your suggestions. Please look at the sentences on lines 74-86, 304-306, 330-337. These changes will allow the reader to better understand the text.
3. The key words section should be after the abstract and not after the methodology section
Thank you. I did
4. The information on the Characterisation of fungi is very important with extremely beautiful pictures to compliment it. Unfortunately, that information does not fall in the methodology section. I will suggest you move these important details to the supplementary section so the manuscript is short and concise. Remember the goal is to keep your readers interested.
Thank you. The characterisation of fungi have been mooved into a separate section.
5. Please ensure the tenses used in the methodology are in accordance with section 4.1.2 of the journal’s guidelines which can be found here https://www.mdpi.com/authors/layout#_bookmark21
The tenses were changed – lines 132-156.
Comments from the previous draft that were not address
Page 6, Line 213: what kind of sieve was used to separate the particle sizes. This information is important to ensure the work can be repeated
Line 114 - 1 mm sieve
Page 6 line 223 to 226
It is not clear how cultures in petri dishes were washed thrice with dd water
Line127 – wash bottle, gentle
It is not clear what kind of filters were used
Line 127 - paper Whatman GT/C
It is not clear how long the samples were dried at 80oC
Line 127 – for 4 hours
Line 120 - Each fungal suspension contained 1.0×109 spores of fungi in 1 ml.
The text (lines 48-63) is very important to the manuscript and links strongly to our manuscript. The text informs the reader about the practical and very effective aspect of use entomopathogenic fungi instead of pesticides in agriculture.
The first sentence of the conclusion (line 340) is fundamental to our manuscript.
The analytical method used in our manuscript (Amino Acid Analyzer) was better than HPLC and GC (table 6). The method is characterized by high specificity, high accuracy and repeatability.
The result of this method was the determination of a larger number of amino acids. This number is significantly larger than in published studies.
Such a number of determined amino acids allowed to present very detailed conclusions. These detailed conclusions are new and of great value to our manuscript. Therefore, I propose to present all these detailed conclusions.